# Interventions on Gut Microbiota for Healthy Aging

**DOI:** 10.3390/cells12010034

**Published:** 2022-12-22

**Authors:** Sabrina Donati Zeppa, Deborah Agostini, Fabio Ferrini, Marco Gervasi, Elena Barbieri, Alessia Bartolacci, Giovanni Piccoli, Roberta Saltarelli, Piero Sestili, Vilberto Stocchi

**Affiliations:** 1Department of Biomolecular Sciences, University of Urbino Carlo Bo, 61029 Urbino, Italy; 2Department of Human Science for Promotion of Quality of Life, Univerity San Raffaele, 00166 Rome, Italy

**Keywords:** gut microbiome, aging, diet, supplements, physical activity, longevity, lifestyle

## Abstract

In recent years, the improvement in health and social conditions has led to an increase in the average lifespan. Since aging is the most important risk factor for the majority of chronic human diseases, the development of therapies and intervention to stop, lessen or even reverse various age-related morbidities is an important target to ameliorate the quality of life of the elderly. The gut microbiota, that is, the complex ecosystem of microorganisms living in the gastrointestinal tract, plays an important role, not yet fully understood, in maintaining the host’s health and homeostasis, influencing metabolic, oxidative and cognitive status; for this reason, it is also named “the forgotten endocrine organ” or “the second brain”. On the other hand, the gut microbiota diversity and richness are affected by unmodifiable factors, such as aging and sex, and modifiable ones, such as diet, pharmacological therapies and lifestyle. In this review, we discuss the changes, mostly disadvantageous, for human health, induced by aging, in microbiota composition and the effects of dietary intervention, of supplementation with probiotics, prebiotics, synbiotics, psychobiotics and antioxidants and of physical exercise. The development of an integrated strategy to implement microbiota health will help in the goal of healthy aging.

## 1. Introduction

Aging is an inevitable biochemical process that results from the body’s limited capacity to regenerate itself. In Western civilizations, the length of life is continually increasing, but aging is frequently accompanied with an elevated risk of chronic illnesses, including cancer and cardiovascular and neurodegenerative disorders [1].

The cellular and molecular indicators for aging are genomic instability, decreased telomere length, epigenetic changes, defective nutrition sensing, loss of proteostasis, cellular senescence, exhaustion of stem cells and altered intercellular communication. These hallmarks of old-age should be taken into account to set anti-aging therapies [2]. Many preventive anti-aging interventions measures, including calorie restriction, dietary intervention, exercise, pharmacological therapies and genetic alterations, have been proposed in several organisms [3,4].

In animal models, the reduction of mammalian targets of rapamycin (mTOR) expression and the increase in sirtuin (SIR) levels through genetic engineering has been demonstrated to prolong lifespan [5].

However, its applicability in human society is hampered by difficulty in managing gene alterations.

Thus, the focus of current anti-aging research is on nongenetic therapies.

In humans, the most promising strategies for improving health and lifespan by lowering the risk of age-related diseases include a correct diet, regular exercise, secure living and working conditions and pharmacological therapy [3,4].

These strategies also influence the biodiversity and functionality of gut microbiota. This “forgotten endocrine organ” works to maintain host health and homeostasis through a delicate balance of commensal and pathogenic bacteria [6].

Although the human host and the gut microbiota are in a dynamic equilibrium, age-dependent exposures can also cause, directly, ecological perturbations of the gut microbiota up to the development of dysbiosis.

On the other hand, a healthy microbiome could directly affect how long people live but, above all, how long people live in good health. In this review, a comparison between the young and old microbiota will be analyzed in order to suggest a putative microbial signature of aging.

The effects of physical activity, dietary interventions and supplements for maintaining good health in aging, with particular attention on the gut microbiome, will be discussed.

## 2. Gut Microbiome and Aging

The gut microbiota is a highly complex and diverse ecosystem of microorganisms living in the gastrointestinal tract [7]. The perfect equilibrium of beneficial and pathogenic bacteria in the gut microbiome contributes to the maintenance of the host’s health and homeostasis, by preserving the intestinal tract’s epithelial integrity, improving food digestion, xenobiotic metabolism and production of several bioactive compounds, stimulating intestinal immunity and limiting the growth of harmful bacteria [8,9].

The gut microbiota diversity and richness change during life: the dynamic nature of microbiota in early life is well recognized, but subtler variations in diversity persist until middle age (about 40 years in humans), at which time there is a period of relative stability [10,11]. This stability is physiologically lost during aging as an aspect of the aging of the prokaryotic and eukaryotic symbionts, also termed the holobionts [12,13,14].

The loss of gut microbial stability leads to a dysbiotic state, characterized by a decrease in the variety of bacterial species and the destruction of helpful bacteria. Increased intestinal permeability caused by microbial polymorphisms can inhibit normal nutrient absorption, food metabolization and immune system regulation [15].

On the other hand, a dysbiotic microbiome can affect the physiopathology of various conditions, including aging; this evidence suggests that host aging and loss of microbial functions are correlated one to one [9].

Numerous studies in animals and humans have shown that the composition of the gut microbiota varies with host age.

Sepp and colleagues analyzed gut microbiota composition of young Estonian people (*n* = 25), born in the late 1990s, and centenarians and oldest-old people (*n* = 25) before the year 1920 [16]. Centenarians and oldest-old people showed a higher richness and diversity of microbiota than young people. *Bacteroides* enterotype was predominant in young people and was characterized by low richness and poor diversity of microbial species, suggesting that this enterotype is more susceptible to external factors. On the other hand, the predominant enterotype in centenarians and oldest-old people was *Prevotella*. The diet of the elderly, rich in potatoes and cereals, certainly had a significant beneficial impact on the microbiota profile.

Centenarians’ microbiota differs functionally from that of young-old adults; for example, they produce more short-chain fatty acids [17,18]. A higher relative abundance of *Cristensenellaceae* in centenarians than in young people has been observed in Italy, China and Korea This abundance is associated with human longevity in general and a normal body mass index and low risk of cardiometabolic disease [19] Furthermore, a high prevalence of Bifidobacterium longum and *Eubacterium coprostanoligenes* in the oldest-old group may lower the risk of this pathology [20].

Many of the changes in the microbiome induced by aging are disadvantageous to human health. An age-related decrease in *Firmicutes* and *Bifidobacteria,* and a parallel enrichment in *Bacteroidetes*, *Proteobacteria* such as *Enterobacteriaceae* have been observed in human studies [14,21,22,23]. These microbial changes induce a low-grade inflammation and immunosenescence—two hallmarks of the aging process. The persistent inflammatory condition, known as inflammaging, is a significant risk factor for morbidity and mortality; chronic infections, inactivity, visceral obesity, nutrition and psychological stress, together with intestinal dysbiosis, are factors that might start and perpetuate inflammaging [24]. The prevention of age-related modifications in gastrointestinal functioning in model organisms reduces microbial dysbiosis and prolongs life [25]. For instance, mucin formation in mice declines with age, resulting in a mucus layer that is thinner and more inconsistent [26]. In the digestive tract, mucin creates a barrier that shields epithelial cells from direct contact with bacteria [27]. Moreover, when this layer is absent, bacteria that normally avoid the epithelial layer might trigger inflammatory reactions [28].

Many bacteria, belonging to *Clostridiaceae*, *Akkermansiaceae*, *Bifidobacteriaceae* and *Bacteroidaceae* families, feed on mucin, but many of these mucin-metabolizing species, such as *Akkermansia muciniphila*, have been found in microbial communities of centenarians, suggesting that this phenomenon can be beneficial (Figure 1) [29]. The supplementation with some strains, *A. muciniphila*, *Bacteriodes fragilis*, *Bacterioides vulgatus, Bifidobacterium* spp. and *Prevotella* spp., can reduce the age-related loss of mucin and have positive effects on the body’s immune system, as well as lengthen the lifespan of progeroid mice [30,31,32,33]. This paradox can be explained by the capacity of *A. muciniphila* and the other strains to both metabolize mucin and increase its synthesis.

Gastrointestinal physiologic changes in aging led to a decrease in colonic transit and an increase in intestinal permeability associated with a local inflammation. Elevated expression of inflammatory cytokines can decrease expression of tight junction proteins (e.g., zonulin and claudins), which increases permeability and could act to perpetuate inflammation [34].

## 3. Gut Microbiome and Cognitive Health

Gut microbiome composition and metabolism are constantly influenced by several factors, such as diet, drug, physical exercise and social environment [21]. In response to these stimuli, the microbial community adapts, changing bacterial composition and function throughout the life of human organisms.

The host-immune system plays a key role in shaping the gut microbiome by allowing commensal bacteria to grow and selectively eradicating or neutralizing pathogenic germs.

Altered intestinal microbiota composition in later life is associated with inflammaging, declining tissue function and increased susceptibility to age-associated chronic diseases, including neurodegenerative dementias.

Parker et al. [35] investigated the hypothesis that modifying the gut microbiota is possible to influence the onset of significant comorbidities related to aging, including inflammation affecting the brain and retina in mice.

The intestinal microbiota of young (3 months), old (18 months) and aged (24 months) mice have been exchanged using fecal microbiota transplantation (FMT) [35].

The transfer of elderly donor microbiota into young mice accelerates age-related changes in the retina, central nervous system and cytokine signaling, as well as the loss of important functional eye proteins. These effects are accompanied by an increase in gut barrier permeability. However, by transferring young donor microbiota, these negative effects can be reversed.

In addition to a number of physical illnesses, cognitive impairment affects a large portion of the aged population. Some centenarians do, however, still have sharp minds, probably correlated to a balanced gut microbiome [36].

The richness and diversity of gut microbiota are acknowledged health biomarkers, including the frailty of elderly people and some personality traits [37,38,39]. Frailer individuals have decreased richness and diversity of gut microbiota [40,41].

In comparison to younger adults, the gut microbiota of the elderly shows more interindividual variability [42,43]. Several factors, such as physiological aging and related comorbidities, mobility, diseases, frailty, polypharmacy and environmental conditions, may be responsible for this significant heterogeneity [18,44,45,46].

Similar results were obtained in previous studies where Italian centenarians had higher diversity and Indian, Japanese and Chinese centenarians had higher richness of gut microbiota than young people in these countries [45,47].

Recently, the role of trimethylamine-N-oxide (TMAO), a metabolite produced by gut microbiota in the aging process, including brain aging, has been investigated by Li et al. [48].

TMAO could induce brain aging and age-related cognitive dysfunction and aggravate the cerebral aging process in progenitors of senescence-resistant strain 1 (SAMR1) and the senescence accelerated mouse prone 8 (SAMP8), which might provide new insight into the effects of intestinal microbiota on the brain aging process and help to delay senescence by regulating intestinal microbial metabolites.

A higher abundance of microorganisms of the environment was found in oldest-old people (Figure 1). Among them, spore-forming gut bacteria such as *Clostridium sensu stricto*, able to regulate serotonin synthesis in the gut, have been identified [49]. Serotonin is a neurotransmitter that modulates mood, cognition, learning, memory and other physiological processes, and it has an important role in the etiology and treatment of depression [50]. Furthermore, another species, *Bacillus licheniformis*, extended longevity in a nematode model. This species may alter aging because it controls the genes linked to serotonin signaling in worms [51].

A key role of healthy longevity could be to have a mind that thinks positively, and the species richness and diversity of the intestinal microorganisms can contribute to this positive mind, suggesting that gut microbiota can be an important marker for well-aging.

## 4. Diet and Aging

According to Kirkwood’s disposable soma theory of aging, organisms age because of an evolutionary trade-off between resources needed for cellular maintenance, development and reproduction because they have limited access to resources [52,53].

Aging is a plastic process that can be influenced by dietary habits. Nutrition affects gene expression and metabolism of the host and gut microbiota, representing a link between the two symbionts (Figure 2) [54].

There are numerous signaling pathways involved in the alteration of gut microbiota and its metabolites, the short-chain fatty acids (SCFAs), which can disturb the host’s normal physiological functions. Significantly, many of these processes happen to be controlled by the mammalian target of rapamycin (mTOR) [55].

The pathway of mTOR regulates cell proliferation, autophagy and apoptosis by participating in multiple signaling pathways in the body and regulates growth and metabolism by promoting protein synthesis in response to nutritional availability, including dietary amino acids [52].

mTOR is an evolutionarily conserved kinase and the catalytic core of two distinct complexes: mTORC1 and mTORC2, defined by the presence of the key accessory proteins Raptor and Rictor, respectively [56].

In particular, mTORC1 activation involves either the Akt/PI3K pathway or the AMPK signaling pathway, whereas the mechanism of mTORC2 is still unclear.

It is interesting to explain the crosstalk between microbiota and the mTOR pathway. Chin et al., 2014, in a previous study, showed that intestinal bacteria products can extend the lifespan as an effect of mTOR inhibition.

The gut microbiota ferments undigested dietary fibers producing SCFAs, which are important in differentiation, growth arrest and apoptosis. The main SCFAs produced are acetic acid (C2), propionic acid (C3), and butyric acid (C4) (ratio 60:20:20) and, in lower amounts, lactic acid (mainly L-lactate), 2-hydroxy propanoic acid, valeric and caproic acids. Acetic and propionic acids are mainly absorbed from the colonic lumen, while butyric acid is the substrate of colonocytes. Inflammatory bowel disease, irritable bowel syndrome, cancer and other pathologies are strictly related to SCFAs production [57].

The beneficial effects of butyrate on cancer, non-alcoholic fatty liver disease and inflammation are mediated through autophagy and apoptosis, although the exact molecular mechanisms are still under research. Probably, mTOR inhibition is involved in this process by the activation of AMPK, which can lead to a formation of autophagosomes [55].

In eukaryotes, the target of the rapamycin (TOR) pathway plays a central role in nutrient and energy sensing. The TOR pathway regulates growth and metabolism by promoting protein synthesis in response to nutritional availability including dietary amino acids [58].

The dysregulation of various nutrient sensing pathways, such as insulin/insulin-like growth factor-1 (IIS), mammalian TOR, (mTOR), AMP-activated protein kinase (AMPK) and SRs, has been linked to an increased risk of non-communicable illnesses with aging [59].

The first nutrient-sensing pathway to be connected to this response is the IIS pathway, which controls glucose homeostasis [60]. IIS pathway downregulation promotes forkhead box O (FOXO) proteins, which increase insulin sensitivity [61], cell cycle arrest [62], and lower inflammation, improve mitochondrial biogenesis and switch the metabolism from glucose to lipid oxidation.

Furthermore, FOXO transcription factors are at the interface of crucial cellular processes, which regulate apoptosis, DNA repair, metabolism, oxidative stress resistance and longevity.

Instead, the c-Jun N-terminal kinase (JNK) signaling pathway is an important genetic factor for lifespan regulation through increased transcriptional activity of FOXO [63].

An innovative study showed experimental drugs that activate FOXO can restore the functionality of geriatric stem cells, enabling an old muscle to activate regenerative processes that are typical of a young muscle [64].

The mTOR kinase pathway detects high concentrations of amino acids [65]. On the other hand, AMPK and SIRs detect nutritional deficiencies compared to IIS and mTOR pathways that identify the abundance of nutrients. By inactivating mTOR and activating peroxisome proliferator-activated receptor-gamma coactivator (PGC-1), respectively, the upregulation of AMPK and SIRs improves the lifespan.

In animal models, dietary or caloric restriction (CR), a moderate reduction in food intake, has been shown to have positive impacts on nutrient sensing pathways and healthy aging [66]. However, strict food regimens in people may not be sustainable. Recently, easier and more practical interventions such as intermittent fasting (IF) have been proposed as alternatives of CR. CR and fasting appear to extend lifespan by downregulating the mTOR signaling cascade, targeting the circadian clock [67].

Using more modest dietary treatments to alter nutrient sensing pathways and support healthy aging has gained popularity [59].

In humans, eating less protein is linked to lower insulin-like growth factor-1 (IGF-1) levels and specific healthful diets are linked to longer telomere lengths, both of which are related to longer longevity.

The association between IGF-1 levels and longevity in humans is very complex, and a recent meta-analysis shows that there is a U-shape nonlinear relationship with mortality; however, bioavailability of IGF-1 is affected by chronic inflammation and diet, and particularly proteins, dietary restrictions, obesity and lifestyle can modulate the level of IGF-1 to a physiological level [68].

In particular, the Mediterranean diet that emphasizes low-to-moderate protein intake, foods with a low glycemic index and foods high in polyphenols could serve as a suitable substitute [69]. As a regulator of both the IIS and mTOR processes, decreasing dietary protein intake modulates circulation IGF-1 levels [70,71,72].

The FOXO transcription factor FOXO3A is then activated by the subsequent downregulation of the IIS pathway, which promotes the transcription of homeostatic genes and reduces the mitogenic effects of RAS (from rat sarcoma viruses) [73].

As shown by the downregulation of phosphorylated mTOR and p70S6 kinase (p70S6K), low IGF-1 concentrations also make it easier for mTOR activity to be inhibited, which in turn lowers cell proliferation. Additionally, it has been proven that Mediterranean diet ingredients such as olive oil, which are high in polyphenols, activate AMPK pathways [74]. This partially stimulates autophagy by blocking the mTOR complex, which has age-protective benefits [75].

The SIR1 also promotes autophagy by upregulating AMPK in a positive feedback loop through acylating and activating LKB1 [76]. Additionally, a diet high in polyphenols causes the interaction of AMPK and SIRs, which leads in the deacylation and inactivation of nuclear factor kappa B (NF-kB), which is probably crucial for the control of immune response and inflammation [77].

A striking relationship between adherence to a Mediterranean (Med) diet and the abundance of fecal SCFAs has also been reported, which strongly correlated with consumption of fruits, vegetables and fiber. This was accompanied with the prevalence of *Prevotella* in plant-based diets, with metagenomic analysis revealing a significant increase in the abundance of genes associated with polysaccharide degradation and SCFA metabolism [78]. In an elderly population (aged 65 years), where subjects were fed low fat and high fiber diets, greater diversity in bacterial microbiota profiles were observed, while those with “moderate to high” levels of fat and “low” fiber intakes had the least diverse microbiota [42]. Collectively, these studies show that intake of dietary fiber influences the diversity of intestinal microbiota and the dominant species, suggesting that a richly diverse microbiota may be beneficial in, for example, protecting against and excluding enteric pathogens that cause intestinal diarrhea-associated diseases.

Furthermore, a low to moderate protein intake is associated with a lower production of trimethylamine N-oxide (TMAO) and *L. ruminococcus* abundance [78].

Long-established habitual dietary habits exert strong control over the makeup of the microbiota, and occasional shifts in dietary habits, such as high (meat) protein intake in vegans/vegetarians, may not lead to the production of harmful metabolites.

Kawano et al. [79] propose an alternative explanation for the pathogenic role of sugar in metabolic disease through suppression of immuno-protective microbiota. These results define a microbiota-dependent mechanism for immuno-pathogenicity of dietary sugar and highlight an elaborate interaction between diet, microbiota and intestinal immunity in regulation of metabolic disorders in aging.

Some evidence suggests that important components of the Med diet, such as the moderate protein consumption, low glycemic index, and abundance of foods rich in fibers and polyphenols, may promote healthy aging through good effects on nutrient sensing pathways, and it is always important to consider the type of macronutrients and micronutrients present in the whole diet rather than just one. The Med diet can greatly affect senescence and related aspects, acting on cellular and molecular hallmarks of aging (Figure 2).

However, other mechanisms remain to be discovered; to this regard, several authors have discovered that modifications to gustatory or olfactory neurons, or even treatment of animals with diet-derived scents, can influence lifespan in *Caenorhabditis elegans* and *Drosophila melanogaster* [54,80,81].

## 5. Supplements

With the increase in evidence directly linking diet and health, several plants and plant extracts (e.g., fruit extracts, leaf extracts, root and tuber extracts) have emerged as possessing potential health benefits (Figure 2).

In general, differences in the composition of the gut microbiota have been reported between older and younger adults; Actinobacteria, particularly *Bifidobacterium* and *Firmicutes*, appear to decline with age, whereas Bacteroidetes and Proteobacteria, particularly *Enterobacteriaceae* and *Clostridia*, appear to increase.

We describe the preventive and therapeutic attributes of phytochemicals such as polyphenols, probiotic microbes and omega-3-fatty acids in influencing the emerging nexus of immunosenescence, cellular senescence and SC during aging. Outstanding questions and nutraceuticals-based pro-longevity and niche research areas have been deliberated. Further research using integrative approaches is recommended for developing nutrition-based holistic immunotherapeutic strategies for ‘healthy aging’.

### 5.1. Probiotics

According to Hill’s definition [82], probiotics are “live microorganisms that, when administered in adequate amounts, confer a health benefit on the host”. Probiotics may be helpful for treating and preventing gastrointestinal diseases [83], irritable bowel syndrome [84], blood pressure [85] and depressive symptoms, according to recent meta-analyses of randomized controlled trials (RCTs) in adult populations [86].

There have been several reports of probiotic strains isolated from elderly individuals; for example, Park et al. [51] selected a probiotic from the feces of a Korean old (over 80) population known for their long life, who had regular bowel movements. The species most commonly found was Lactobacillus fermentum, and, in particular, one isolate demonstrated the strongest adhesion to intestinal epithelial cells, the greatest immune-enhancement, anti-inflammation and anti-oxidation activity, as well as the greatest survival rates in the presence of synthetic gastric juice and bile solution. This isolate, known as *Lactobacillus fermentum* PL9988, possesses every quality needed in a probiotic. More recently, Fang et al. [87] selected four bacteria from centenarians (*L. fermentum* SX-0718, *L. casei* SX-1107, *B. longum* SX-1326 and *B. animalis* SX-0582) and tested their combination as a probiotic in SAMP8 model, evaluating behavior, neuroinflammation, intestinal inflammation and microbiota composition. In aged mice, this probiotic combination reduced motor dysfunction, diminished exploratory behavior and impaired spatial memory. The regulation of gut microbiota and inhibition of TLR4/NF-kB-induced inflammation may be how the probiotic combo has anti-aging benefits.

Because immune function declines with advancing age, Miller et al. performed a meta-analysis of controlled studies showing that the tumoricidal activity of natural killer (NK) cells and the phagocytic capacity of polymorphonuclear cells increased after brief probiotic supplementation in the elderly [80], compared with controls, even though the results obtained in the different studies were heterogeneous.

The effects of probiotics on inflammaging have been tested; for example, Lefevre et al. [88] investigated the effects of *B. subtilis* CU1 on immune system of the elderly, showing increased fecal and salivary secretory IgA concentrations compared to the placebo, with decreased frequency of respiratory infections in the winter period.

Recently the effects of a probiotic diet on the well-being of healthy seniors living in boarding and private homes were analyzed after a 6-month treatment [89], showing an increase in fecal Lactobacilli and Bifidobacteria and a decrease in Proteobacteria. Actinobacteria and Verrucomicrobia phyla augmented this, especially due to the *Akkermansiaceae* and *Bifidobacteriaceae* contribution at the family level.

Furthermore, SCFAs and butyric acid were significantly higher in the probiotic group, with anti-inflammaging effects.

Probiotic treatment can promote interactions between key constituents of the microbiota and the host epithelium, modulating transcriptional response of the gut microbiota [90]. Aging is also associated with an increase in opportunistic pathogens, sometimes linked to antibiotic use. Clostridium difficile, recently reclassified as *Clostridioides* difficile [91], is often associated with diarrhea and subsequent inflammation and nutrition problems and a reduction on *Bifidobacteria* [92]. A combination of probiotics was shown to reduce the probability of *C. difficile* inflammation among elderly patients who undergo proximal femoral fracture surgery [93].

It should be noticed that studies on probiotic supplementation might lead to conflicting results, since several parameters, such as dosage, strains and duration of administration, need to be controlled. Furthermore, host-dependent factors, such as age, disease, drug use (especially antibiotics), diet and lifestyle, can influence outcomes.

### 5.2. Prebiotics

Gibson defined prebiotics as “non-digestible (by the host) food ingredients that have a beneficial effect through their selective metabolism in the intestinal tract” [94]. Prebiotics should fulfill three criteria: “(a) resistance to gastric acidity, hydrolysis by mammalian enzymes and gastrointestinal absorption; (b) fermentation by intestinal microbiota; (c) selective stimulation of the growth and/or activity of intestinal bacteria associated with health and wellbeing”.

Gut bacteria use dietary fiber and non-digestible oligosaccharides as fuel, producing short-chain fatty acids playing a role in the regulation of cellular processes. Prebiotics specifically stimulate the growth of endogenous bacteria such as *Bifidobacteria* and *Lactobacilli* [95]. For this reason, prebiotics can help in protecting the immunological system through microbiota action [96].

Chung et al. [97] evaluated the effects of xylooligosaccharides (XOSs) on the intestinal microbiota, gastrointestinal function and nutritional parameters in subjects over 65 years of age. In the treatment group, supplemented with 4 g of XOS per day for 3 weeks, the population of *Bifidobacteria* and the fecal moisture content significantly increased, while the fecal pH value decreased. The nutrient intakes, GI function, and blood parameters were not modified, showing no adverse effects on nutritional status in the elderly.

Galacto-oligosaccharides (GOS) can reverse the age-related decline in *Bifidobacteria*, modulating associated health parameters. Vulevic et al. [98] analyzed the effect of GOS mixture (Bimuno (B-GOS)) on gut microbiota and immune function in 65–80-year-old subjects. B-GOS consumption led to significant increases in Bacteroides and *Bifidobacteria* and in an improvement of immune function (higher Interleukins-10 and 8, natural killer cell activity and C-reactive protein and lower Interleukin-1β).

Scheid et al. [99] investigated the effect of 9 weeks of daily intake of 7.4 g of fructooligosaccharides (FOS) as freeze-dried powdered yacon on glucose, lipid metabolism and intestinal transit in a group of elderly people. The prebiotic consumption did not adversely affect intestinal transit and led to a decrease in serum glucose but not lipids.

Inulin is a soluble dietary fiber, promoting growth of bacteria that metabolize it to SCFAs; the capacity of SCFAs production and host response in mice of different ages was analyzed [100]. Mice across young (5 months), middle (11 months) and old (26 months) age were subjected to 6 weeks of an inulin-containing diet. Positive inulin-induced changes were more effective in middle-aged mice than old ones, suggesting that age should be considered when analyzing supplementation effects. Further studies on inulin effects in older humans are needed.

Aging is associated with loss of skeletal muscle mass and strength, leading to sarcopenia and frailty. A study conducted on subjects over 65 years old andliving in nursing homes showed a protective effect of a prebiotic formulation (inulin and FOS) on frailty index levels [101].

This effect was more evident in subjects with higher levels of frailty. It is known that older people present an ‘anabolic resistance’, that is, the inability to anabolize muscle in response to protein supplementation. The modulation of the gut microbiome using a prebiotic, in addition to protein supplementation, can improve muscle strength supplementation alone [102], suggesting new aids to mitigate sarcopenia and frailty in older adults.

### 5.3. Ω-3 Fatty Acids

Ω-3 fatty acids (FAs) are polyunsaturated FAs (PUFAs), including docosahexaenoic acid (DHA), eicosapentaenoic acid and docosapentaenoic acid, mainly contained in fish meat, eggs, seafood and vegetable oil [103]. Ω-3 fatty acids have been linked to improvements in the composition and diversity of the gut microbiome in middle-aged and elderly women [104], and the concurrent administration of ω-3 fatty acids and probiotic strains provides amplified health benefits [105]. Ω-3 fatty acids increase levels of LPS-suppressing bacteria (i.e., *Bifidobacteria*) while decreasing levels of LPS-producing bacteria (i.e., Enterobacteria); furthermore, they interact with the *Firmicutes/Bacteroidetes* ratio, increasing *Lachnospiraceae* taxa, and thus promoting the production of the SCFA butyrate [106]. EPA and DHA have also been shown to positively influence the gut microbiota composition by supporting a lean phenotype [107].

Furthermore, they possess anti-inflammatory properties and prevent leaky gut and positively modulate host microbial ecosystems [108], and the presence of ω-3 FAs, primarily DHAs, in brain lipids is thought to be important for maintaining brain structure and function as well as cognitive health [109].

### 5.4. Synbiotics

As already reported in the previous paragraphs, the effect of probiotic administration varies according to factors such as sex, age and lifestyle, and often the selected strains are found in fecal samples for a limited time. Prebiotics are substances able to support specific bacterial strains, favoring their engraftment and permanence in the intestine, providing useful products such as SCFAs. For this reason, probiotics and prebiotics working synergistically are often administered in a single product. The International Scientific Association for Probiotics and Prebiotics (ISAPP) in 2019 updated the definition of a synbiotic to “a mixture comprising live microorganisms and substrate(s) selectively utilized by host microorganisms that confers a health benefit on the host”, emphasizing that a synergistic synbiotic is a synbiotic for which the substrate is designed to be selectively utilized by the co-administered microorganisms [110].

They manifest their action throughout several physiological (and occasionally pathological) processes, such as tissue remodeling, cancer, injury and aging and function as a cellular defense mechanism to prevent cell damage. Additionally, gut barrier leaks that allow bacteria and/or microbial components to infiltrate may trigger inflammatory reactions [111].

The results of a study performed in healthy elderly subjects suggest that consumption of lactitol combined with L. acidophilus NCFM may improve some markers of the intestinal microbiota composition and mucosal functions [112]. Recently Cicero et al. [113] analyzed 60 elderly patients following a standardized diet and training protocol treated with a synbiotic formula of *L. plantarum* PBS067, *L. acidophilus* PBS066 and *L reuter*i PBS072 with active prebiotics or placebo. The first ones experienced a statistically significant improvement in waist circumference and in fasting plasma insulin, total cholesterol, high-density lipoprotein cholesterol, non-HDL-C, triglycerides (TG), low-density lipoprotein cholesterol, high-sensitivity C-reactive protein and tumor necrosis factor alpha serum levels, compared both to the baseline and the control group. A significantly greater improvement was observed also for visceral adiposity index, mean arterial pressure, TG and fasting plasma glucose.

Cellular senescence is a feature of aging and is just now being studied in relation to the gut. The senescence-associated secretory phenotype, which is caused by an accumulation of senescent cells, can have harmful effects on the tissue environment, even though these cells are advantageous when they are present acutely. Such signaling pathways may contribute to systemic metabolic dysregulation and a number of age-related diseases by promoting inflammaging and tissue malfunction. Probiotics and their synbiotic amalgamation with plant polyphenols should be studied in the context of cellular senescence, which may ultimately help devise probiotic-based anti-senescence strategies [114]. Pro-longevity abilities of a synbiotic supplement made of the polyphenol-rich ayurvedic herb Triphala and a probiotic blend including *L. plantarum, B. longum* spp. infantis and *L. fermentum* demonstrated in Drosophila through a decrease in age-related physiological stress, oxidative stress and low-grade inflammation [115]. Although it is not possible to translate the results obtained in a fruit fly to humans, this strategy needs further study.

### 5.5. Brain Health and Psychobiotics

The onset of depression and neurodegenerative diseases often linked to altered metabolic conditions is frequent in the elderly.

The etiology of neuropsychiatric illnesses and the regulation of brain processes are both thought to be influenced by the gut microbiota, according to evidence. The gut bacteria and brain can communicate with each other continuously in both directions through a variety of ways. An association between oxidative stress and the pathophysiology of major depressive disorder via the hyperactivation of inflammatory responses has been suggested by recent research, which emphasizes the interexchange of signals influenced by the gut microbiota that are detected and transduced in information from the gut to the nervous system involving neural, endocrine and inflammatory mechanisms. Donati Zeppa et al. recently analyzed in a review the link between microbiota and depression and on possible strategies to counter this condition [116].

Dysbiosis of the gut microbiota was detected in Alzheimer’s disease (AD) [117], since comparing AD patients with non-AD there are in the first one decreased levels of Bacteroidaceae, Veillonellaceae and Lachnospiraceae, and increased levels of *Ruminococcaceae, Enterococcaceae* and *Lactobacillaceae*. The production of amyloid proteins by a number of potentially harmful gut inhabitants has been suggested as a possible precursor to pathological amyloid protein misfolding [118], and in Parkinson’s disease, where a microbiological pattern of gut microflora that can trigger local inflammation and subsequent aggregation of α-synuclein and generation of Lewy bodies has been detected [119].

It has been demonstrated that bacteria produce neurotransmitters, such as gamma-aminobutyric acid, acetylcholine and serotonin, encourage the production of serotonin by gut epithelial cells, produce bioactive substances and SCFAs, and release metabolites that could cross the blood brain barrier and enter the bloodstream [120]. Furthermore, gut bacteria can control the expression of central neurotransmitters and associated receptors [121].

Through primarily the vagus nerve, the enteric nervous system (ENS) serves as a conduit for the two-way communication between the brain and the gut, and Santos et al. well-summarized the bidirectional pathway linking PD to gut [122]

In 2013, Dinan et al. defined psychobiotic as “a live organism that, when ingested in adequate amounts, produces a health benefit in patients suffering from psychiatric illness” since they produce and deliver neuroactive substances such as gamma-aminobutyric acid and serotonin, which act on the brain–gut axis [123]. In 2016, Sarkar et al. expanded this definition to encompass prebiotics, which enhance the growth of beneficial gut bacteria [124]. Several studies demonstrated that the effects of psychobiotic supplementation of probiotics containing *L. acidophilus*, *L. casei*, *B. bifidum*, and *L. fermentum* for 12 weeks positively affect cognitive function and some metabolic statuses in AD patients [125].

Patterson et al. [126] reported that intervention with GABA-producing lactobacilli has the potential to improve metabolic and depressive-like behavioral abnormalities associated with metabolic syndrome in mice, suggesting potential therapeutic implications in humans.

Kimura-Todani et al. [127] analyzed the effect of special diets containing acetylated starches that can reach the colon without being absorbed in the upper gastrointestinal tract of male mice, acetate, butyrate and propionate in the cecum, exerting anxiolytic effects on behavioral phenotypes of the host.

A prebiotic-mediated proliferation of gut microbiota in rats, such as probiotics, has been shown to increase central brain derived neurotrophic factor expression, possibly through the involvement of gut hormones [128,129].

In humans, four-week intake of a fermented milk product with probiotic (FMPP) by healthy women affected activity of brain regions that control central processing of emotion and sensation for 4 weeks [130]. Smith et al. [131] examined the acute effects of oligofructose-enriched inulin (5 g) over a 4 h period during which the participants remained in the laboratory, showing improvement in performance and mood tasks, above all on the episodic memory tasks where consumption of inulin was associated with greater accuracy on a recognition memory task, and improved recall performance (immediate and delayed).

Recently Gualtieri et al. reported the effect of probiotic administration in adults carrying an interleukin-1β polymorphism related to high cytokine levels that potentially affects mood disorders, reporting a decrease in anxiety symptoms and suggesting the importance of genetic association studies for psychobiotic-personalized therapy [132].

Additional human research using prebiotic, probiotic and synbiotic therapies to achieve therapeutic manipulation of the microbiota-gut–brain axis are needed, and older people would greatly benefit from this increase in knowledge.

### 5.6. Antioxidant

As already well known, a dietary model rich in polyphenols or integration with polyphenol supplements can help to counteract the dysbiosis [133] of the gut microbiota linked to age but, primarily, to modulate oxidative stress to improve intestinal permeability [134], which is closely associated with chronic activation of the inflammatory response [135]

#### 5.6.1. Resveratrol

Resveratrol is a natural polyphenol present in several foods that has a wide spectrum of pharmacological properties for the management of diabetes, cardiovascular and neurological diseases, and can be considered a multi-target therapeutic agent for chronic diseases [136,137,138].

As already widely discussed in the literature, resveratrol can lead a change directly on the composition of the intestinal microbiota by inhibiting the growth of individual microbial species or causing the transfer of the bacterial population [139]. In turn, intestinal microorganisms can improve the bioavailability of resveratrol, helping the metabolism of resveratrol precursors to resveratrol [140], suggesting a two-way interaction between resveratrol and gut microbiota.

Furthermore, some beneficial effects of resveratrol are associated with gut microbes, including a reduction of the *Firmicutes:Bacteroidetes* ratio and promoting the diversity of microbiota. For example, the administration of resveratrol causes *Enterococcus faecalis* to grow and increased abundance of *Lactobacillus* and *Bifidobacterium* [141].

Other studies have also confirmed that resveratrol can enhance microbial diversity and intestinal barrier function, having a critical role in the fight against obesity [142,143].

Moreover, animal model studies [144] show that resveratrol leads to an attenuation of TMAO, through a modulation of the gut microbiota and increasing the relative abundance of *Bacteroides*, *Lactobacillus*, *Bifidobacterium* and *Akkermansia*, useful for treating atherosclerosis.

These strains have been noted to be found in the gut microbiota of healthy elderly subjects, thus assuming that the increase in the population of these bacteria leads to several improvements [145,146].

Furthermore, resveratrol, decreasing the oxidative stress, is a potential treatment of intestinal diseases [147].

Wellman et al. demonstrated that the disruption of SIRT1, in mice, induces intestinal inflammation by regulating the gut microbiota [129], suggesting that intestinal SIRT1 might therefore be an important mediator of host–microbiome interactions.

As already widely known, resveratrol is one of the agents that activates SIRT1 in animal and human studies [148,149].

#### 5.6.2. Flavonoids

Flavonoids are a class of polyphenols present in different plants (*Ginkgo*, *Calendula officinalis*, *Vitis vinifera*, *Rhododendron nivale* Hook).

An important source of flavonoids is *Rhododendron nivale* Hook (R.n), a plant with anti-aging pharmacological activity, used in Tibetan medicine to delay aging [150].

More than the original structure of flavonoids per se, the positive facilitative effect of flavonoids on organisms can be attributed to the formation of phenolic metabolites in small molecules by colon metabolism. Moreover, the unmetabolized flavonoids have the same ability to remodel the gut microbiota, which, in turn, has a corresponding effect on flavonoid absorption in the gut [151,152].

The interactions of flavonoids with the gut microbiota are already well known, but a further study of flavonoids present in R.n has been carried out by Guo and colleagues, focusing on anti-aging properties [150].

To clarify these anti-aging mechanisms, intestinal microflora and plasma metabolomics, together with bioinformatics analysis were evaluated in animal models.

In particular, the pharmacological study of the ethanol extract components of R.n, such as Myricetin-3-β-D-xylopyranoside, hyperin, goospetin-8-methyl ether 3-β-D-galactoside and diplomorphanin B, revealed that the target function of the active ingredients and pathway enrichment mainly focused on the antioxidant system of glutathione.

The results showed that R.n flavonoids reshaped the disordered intestinal microorganisms and attenuated D-galactose-mediated decline in glutathione oxidase expression, further confirming that the anti-aging effect of flavonoids is closely related to the regulation of the antioxidant system of glutathione [150].

#### 5.6.3. Curcumin

Curcumin has been reported to have many beneficial health properties ranging from anti-aging, anti-cancer, anti-hypertensive, anti-inflammatory and anti-neurological effects. The anti-aging properties of curcumin can be attributable to its ability to act on Sirtuin [137].

In a study by Grabowska and colleagues, different doses of curcumin were evaluated, and it was observed that low doses of curcumin could increase levels of SIRs [153]. In addition, the ability of curcumin to alter mitochondrial uncoupling protein 2 (UCP2) in rats has been observed, since this protein plays a critical role in regulating ROS production.

It has been determined that curcumin in the diet of young and old rats reduces the production of ROS in young UCP2 but not in UCP2 knockout (UCP2-/-) aging mice. Curcumin also restores dependent vasorelaxation cerebrovascular endothelium that was impaired in older rats, demonstrating that curcumin could improve cerebrovascular dysfunction that occurs during aging [154,155,156].

As already widely known, microbiota can be responsible for drug metabolism and bioavailability. The microbiome changes over the course of a lifetime [14], and aging is associated with a reduction of microbial diversity in terms of composition, quality and quantity. It is suggested that healthy aging correlates with microbiome diversity [157]. There is some indication that curcumin is able to modulate gut microbial composition (i.e., biodiversity) [158,159,160], reducing some negative consequences of aging.

In summary, the impacts on the microbiota of curcumin could positively affect the microbiota.

In addition, the bi-directional interaction between the two causes the microbiota to metabolize curcumin into active metabolites, can improve the bioavailability of curcumin and, thus, increase the beneficial impact on the microbiota.

## 6. Exercise

As we age, our microbiota composition is affected by our diets, habits and other aspects of our lifestyle. There is growing interest in understanding how we can maintain a healthy microbial community in our gut or restore a dysbiotic one and the relationship of the gut microbiota to overall health. Lifestyle factors, including physical activity and dietary habits, may affect skeletal muscles and immune aging positively [161]. In particular, the role of exercise in maintaining health has been thoroughly studied, and based on several cross-sectional studies, it appears likely that athletes harbor distinct microbiome compositions compared to less active individuals [162,163].

In this regard, variations in gut microbiota compositions have been observed in subjects according to the amount of physical activity that they do or their fitness levels, indicating that physical exercise may have a beneficial effect on the gut microbiota [164]. On the other hand, the high stress induced by an excessive volume of endurance training [165,166], during which athletes train for four to six hours a day, six days a week, for several weeks without taking breaks from this intense training, has been shown to have negative effects on the gut microbiota. In addition, more recently, Morishima So et al. speculated that the higher abundance of inflammation-related bacteria and higher concentration of succinate found in a group of female elite endurance athletes could result in a form of dysbiosis [167].

The alterations in the gut microbiota produced by excessive intense exercise can also produce gastrointestinal disorders [167].

Indeed, it has been demonstrated that exercise to exhaustion can alter the balance between the gut microbiota and the immune system [168] and that the increased inflammation levels or gastrointestinal damage can adversely affect athletic routines that, in some cases, may result in withdrawal from competitions [169].

On the other hand, more moderate exercise is thought to be beneficial to the gut microbiota [170,171], supporting butyrate-producing bacteria. Indeed, concentrations of butyrate were found to increase in humans after physical exercise [172,173].

Moreover, the microbiota of women with active lifestyles were found to differ from those with a more sedentary lifestyles, with an enhanced presence in the former of potentially beneficial bacteria, such as *Bifidobacterium* spp., *Roseburia hominis*, *A. mucinphila* and *F. prausnitzii* [164].

The positive impact of moderate to vigorous physical activity on the gut microbiota has been supported by several studies that have identified particular microbiota associated with individuals engaging in different levels of physical activity [174]. In this regard, different microbiota compositions were found in professional athletes and healthy controls, with the former showing higher bacterial richness [6,162]. Surprisingly, athletes show fewer *Bacteroidetes* and more *Firmicutes. A. muciniphila* increased both in athletes and in low BMI controls.

However, we must be careful when comparing athletes and non-athletes since athletes are often selected from within small groups of people who already have particular characteristics in terms of high levels of strength (large muscle mass) and high levels of cardiovascular fitness but also a great predisposition to withstand particularly stressful situations. In addition, athletes are individuals who, from a very young age, have lifestyles characterized by behaviors that lead to optimal health, including good dietary habits, which play a very important role. On the other hand, when non-athletes or sedentary subjects are the object of investigations, the effects of exercise interventions have not always yielded significant results. Hence, we must seek to understand the type of physical activity prescribed, what the optimal frequency of weekly sessions should be and how long each session should last to achieve a healthier microbiota.

Interestingly, although a healthy gut microbiota has been associated with good fitness levels and an unhealthy one with cardiometabolic risk factors in adults, the effects of exercise interventions on the gut microbiota of elderly subjects are unclear.

In this regard, a study by Taniguchi et al. [175] that assessed the effects of a five-week endurance training program on a cohort of men aged 62 to 76 years suggested that short-term endurance exercise does not appreciably influence diversity and composition of gut microbiota in this older age group. Likewise, Munukka et al. [176] observed that a six-week endurance exercise intervention only modestly changed the overall composition of the microbial community in 18 overweight female subjects. Taking a different approach, Cronin et al. [177] investigated whether eight weeks of a combined (aerobic and resistance) exercise program, with or without whey protein supplementation, could alter the composition and function of the gut microbial community in a group of primarily overweight or obese male and female adults (*n* = 90; age 18–40). Subjects who were randomly assigned to the exercise groups had to do three sessions a week of progressive resistance training and moderate-intensity aerobic training (18 to 32 min in length). Compared to the baseline, neither exercise group’s post-intervention analysis of taxonomic makeup nor the analysis of metabolic pathways revealed any appreciable alterations. In contrast with the group that received whey protein alone, a trend towards greater bacterial diversity was observed in the exercise group. Small changes in microbial metabolism were only partially reflected by metabolomic and metagenomic investigations. Despite the relatively high sample size of the study, the authors pointed out that self-reported maintenance of typical dietary intake and a wide BMI range may have hampered detection of more substantial changes. The literature on exercise-induced microbiome changes lacks studies comparing different exercise modes and their relationship with gut microbiome composition [167].

The metabolic requirement for energy varies according to the mode of exercise, namely whether it is aerobic or anaerobic. In addition, the metabolic pathway and fuel source used may impact the response of the human gut microbiome. Mailing et al. suggested that future inquiries should be designed to examine changes in gut microbiota related to other modalities of exercise, such as resistance or combined training [178]

In this regard, Dierdra Bycura et al. [179] examined changes to the human gut microbiome resulting from an eight-week intervention of either cardiorespiratory exercise or resistance training exercise performed three times per week. The authors observed no significant changes during or after the exercise intervention in individuals who had been assigned to the resistance exercise program. On the other hand, the subjects assigned to the cardiorespiratory exercise program showed a compositional shift in their gut microbiome by the second week of the exercise intervention. These findings are noteworthy; however, further studies are needed before it can be concluded that resistance training does not affect gut microbiota.

Interestingly, when aerobic and resistance exercises are combined, the results are different. Research by Fei Zhong et al. [37] showed that sedentary elderly women may exhibit partial alterations in the relative abundance of their gut microbiota and community structure after undergoing a combined aerobic and resistance exercise program four times per week for eight weeks. In addition, it was found that exercise can decrease the abundance of pro-inflammation-related bacteria such as Proteobacteria and increase the abundance of anti-inflammation-related bacteria such as *Verrucomicrobia* [37].

To date, although there seems to be growing evidence that exercise can affect the gut microbiota even in the elderly, the optimal amount and mode of exercise that can yield beneficial effects on the gut microbiota in this population have yet to be determined. To this end, the effects of exercise frequency on gut microbiota in the elderly were studied by Qiwei Zhu et al. [180] on a sample of 897 elderly (61- to 70-year-old) and 1589 adult individuals (18–60 years old). To study the impact of exercise frequency on microbial diversity, the elderly subjects were separated into five groups: never, seldom (a few times/month), sometimes (1–2 times/week), regularly (3–5 times/week), and daily. The findings showed that microbial diversity increased with exercise. Daily exercise resulted in a gut microbiota composition that was similar to that of adults aged 18 to 60, and regular exercise enhanced the relative abundance of bacterial functional pathways involved in nucleotide metabolism, glucose metabolism and lipid metabolism. Additionally, researchers found that the microbial diversity and composition of overweight older people significantly changed, which led to changes in the functional pathways used by bacteria for glucose, nucleotide and vitamin metabolism. Furthermore, frequent exercise changed the microbial composition in several ways. Overall, these results demonstrate that regular exercise helps overweight older people to lose weight and suggests that regular exercise may play a role in preserving the integrity of the gut microbiota in elderly people [170].

Regular exercise appears to have favorable effects on gut microbiota in older people, even after just a short period of time—roughly two months. Hence, exercising regularly over the long term may preserve the health of the intestinal microbiota, which can be linked to a higher level of well-being in subjects who exercise regularly (Figure 2).

In this regard, in a large-scale study involving 2051 subjects aged 51–89, Shi J et al. [181] found a relationship between regular exercise over a period of 10–15 years and gut bacterial B-diversity, particularly among women. When compared to participants who did not exercise, those who did showed significantly less *Ruminococcus* abundance. Moreover, *Prevotella* and *Coprococcus* were more prevalent and *Parabacteroides* and *Ruminococcus* were less prevalent in the groups of subjects who engaged in regular long-term exercise. People over 75 who exercised regularly showed the most noticeable increase in *Coprococcus* prevalence. As a result of long-term regular exercise, the investigators found a significant decrease in *Ruminococcus* abundance and a significant relationship between regular exercise and β diversity in the gut microbiota. Men and women who exercised also consumed more dietary fiber, had lower smoking rates and were less likely to be regular drinkers [182]. Whether men or women engaged in regular exercise did not affect BMI or total calorie intake.

To date, the literature shows that physical activity, in particular endurance training, if performed excessively, can potentially lead to a condition of dysbiosis. On the other hand, most athletes have a healthier microbiota than their sedentary counterparts. Indeed, physical activity, when performed in adulthood or old age at moderate or vigorous intensity, improves the health of the gut microbiota while also improving the metabolism of sugars. However, unlike cardiorespiratory exercise, resistance exercise does not seem to have positive effects on the gut microbiota in the medium-to-short term. On the other hand, combining aerobic exercise and resistance exercise seems to stimulate healthy microbiota adaptations. The first positive adaptations in adults and the elderly can be observed after at least eight weeks of training, while the best adaptations are obtained with an exercise frequency of at least three to five times per week. Finally, in conclusion, to maintain a healthy gut microbiota we should stay physically active for as long as it is possible well into our twilight years.

## 7. Conclusions

Despite numerous interindividual differences, it is now clear that the composition of the microbiota of the elderly differs significantly from that of young and middle-aged people. Numerous progresses have been made on the study of the microbiota, and evidence is accumulating on the efficacy of therapies based on the microbiota, even if the clinical applications on obtaining the slowing down of aging are still lacking and more advanced human studies at strain-level resolution are required. The gut microbiota research should be pointed on specific signatures related to longevity. Future research should consider the individuals’ baseline microbiome features and customize the therapies to meet their needs. Such studies’ findings should offer the scientific cues for developing strategies for particular age groups of adults. Predictive models based on machine learning could help in predicting the individual response to specific interventions with different duration and dosage, according to baseline host characteristics and in evaluating how well these relationships translate across various subpopulations. Furthermore, models of accelerated human aging such as progeria and Down syndrome can be useful, instead of lifelong human investigations. Although knowledge on the microbiota in humans is limited, the present evidence led to hypothesizing strategies useful for maintaining a good state of health in the elderly.

A healthy lifestyle, with a balanced diet rich in unrefined foods of natural origin, together with adequate physical exercise, aerobic or combined, sustained for sufficiently long periods, allows for restoration and maintenance of a healthy microbiota even in old age, promoting healthy aging.

Of course, prevention of the decline of the microbiota with a delayed onset of age-related pathologies should be preferred; however, a detailed knowledge of the actions of gut microorganisms will allow the formulation and diffusion of products containing supplements such as probiotics, prebiotics and nutraceuticals with specific properties. Obviously, the use of supplements must be targeted, individualized and calibrated on the needs of the individual subject, and appropriate strategies must be implemented to maintain the restored microbiota.

In this review, we focalized on the influence of lifestyle on the maintenance of a healthy microbiota in the elderly and on the consequences on the general state of health of the subject. The effects of specific supplements were also highlighted, in order to suggest personalized microbiota-based strategies for healthy aging.

Translating knowledge of the microbiome for clinical benefit in the elderly is a difficult challenge for scientific researchers, taking into account the complexity of factors such as diet, environment, stress and drugs that can influence human biology, but also an inspiring opportunity to discover the keys of healthy old age and making them usable as a tool for people who want to live long and healthy.

## Figures and Tables

**Figure 1 cells-12-00034-f001:**
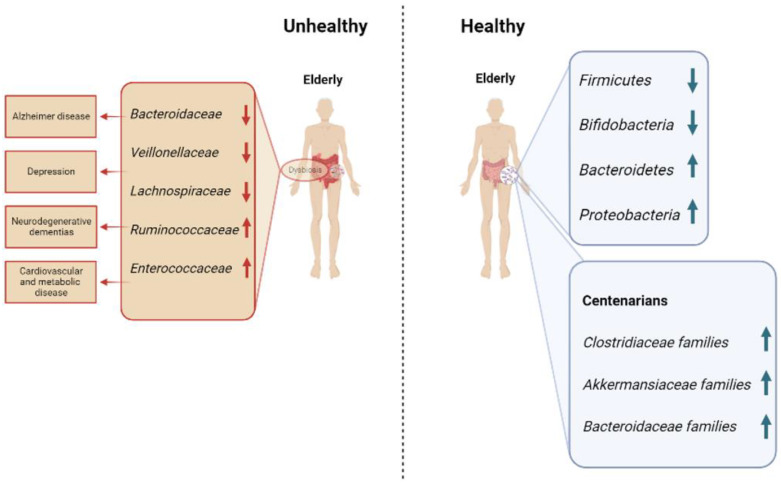
Microbial communities commonly found in unhealthy and healthy elderly, with a focus on centenarians.

**Figure 2 cells-12-00034-f002:**
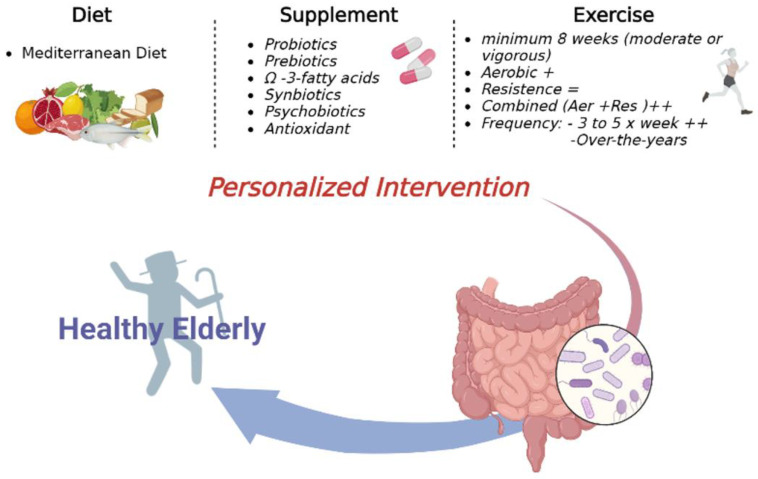
Personalized Interventions on microbiota to support healthy aging.

## Data Availability

Data sharing not applicable.

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
