# Peer review of "Interventions on Gut Microbiota for Healthy Aging"

_cells, 2022, doi:10.3390/cells12010034_

Round 1

Reviewer 1 Report

l like the paper as it is and feel that it will be well accepted by scientific community. The subject seems very important and opens a window for the future studies.

There are couple remarks

1. In the figure 1 protobacteria need to be changed into proteobacteria. In the figure legend it is necessary to explain who are the "pathobionts"/ The term looks too general.

2. C.difficile is an old definition which was changed 6 years ago. I suggest that it is sufficient time for the public to get used for new classification. I understand that the public is not ready for new names of bacteroids or firmicutes.

3.There is one problem which need to be explained or at least noted. Lactobacillus which is more abandoned in case of pathological aging ( fig1) is suggested to use as probiotic for improving the health condition of unhealthy elderly. I think it need to be discussed.

4. In lines 84-85 authors wrote ".... Bacteroidetes, Proteobacteria and pathobionts such as Enterobacteriaceae have been observed in human..."

I can see two mistakes here: some phylum proteobacteria includes Enterobacteriaceae as well as many pathobionts. I suggest to make the appropriate changes  

Reviewer 2 Report

This manuscript presented a wide-ranging discussion on the subjects of health, gut microbiota, and aging. The authors did a fine job putting this review article together, categorizing and organizing the numerous research into 5 topics, and synthesizing a cohesive picture of the current status of gut microbiota and aging research. Here, I share several comments and concerns regarding this manuscript. Hopefully, these suggestions can help improve this manuscript.

  1. Overall, this is an information-rich, data-poor, conclusion-less review. For example, in Sections 2 and 3, the authors brought out a lot of ideas and concepts but not enough detail about the studies used to support the effects indicated. A bit more detail about the experimental design, the models used, and the statistical power of the results and conclusions from each study is necessary for the audience to gauge the validity and significance of the literature cited. In most cases, gut microbiota studies do not lead to a “yes or no” answer, and the conclusion may only apply to a specific model or population group. Therefore, the context of the studies cited is important. Section 5 did a much better job on this front.
  2. Some statements in this manuscript are more speculative than factual, such as in Line 132 “It's possible that these centenarians understand how important a balanced gut microbiome is for brain health [31].” The idea of the gut microbiome is not even two decades old, and it is ambiguous what “these centenarians understand” means in this sentence. Similar concerns also apply to “personality traits”, “social networks”, and “social interactions”. It is hard to prove or disprove the correlations or causal relations between these factors to the specific changes in gut microbiota. Overall, the studies and observations regarding centenarians and social factors suffer from a small sample size and no or weak statistical significance.
  3. The title and Introduction both emphasized that aging and gut microbiota are the focus of this manuscript. However, the current state of gut microbiota research and methodologies may have a hard time resolving all the changes as well as their biological significance over the course of aging. Based on the information presented in this manuscript, it is really a comparison between the young and old microbiota, the before and after, rather than a time course.
  4. A review article on the intersection of two subjects is a difficult feat. Information needs to be integrated rather than piled together. For example, between lines 176-240, there is no mention of gut microbiota. Please correlate the pathways and genes to how gut microbiota is involved in these pathways. Also, in many sections and paragraphs, the interactions seemed not between the gut microbiota and aging, but rather diet, exercise, and host physiology are the driving force for both gut microbiota changes and aging. The limited number of studies that investigated the mechanism of action and the inability to pick apart the causal relationship between changes in microbiota and other endpoints are the most critical issues.
  5. Last but most importantly, what is the definition of a “good” or “healthy” gut microbiota? Inferred from this manuscript and most literature, higher diversity and richness, more Firmicutes and Bifidobacteria, and less Bacteroidetes and Enterobacteriaceae, are considered “good”. Since it is not well established that the microbiota genera and species correlate strongly with the functions, the current understanding of the microbiota changes, using metagenomic techniques, cannot provide enough information to support such conclusions. The genera and species that researchers focus on in most studies overlap, and a lot of different interventions lead to similar changes in the diversity, richness, and compositions of the gut microbiota. I am wondering if the authors can leverage their knowledge and understanding after reviewing so many papers, to point to a direction, or give some suggestions for the gut microbiota research field, to collectively improve the quality of the research.
